# Analysis of Wear Properties of Hardox Steels in Different Soil Conditions

**DOI:** 10.3390/ma15217622

**Published:** 2022-10-30

**Authors:** Krzysztof Ligier, Martyna Zemlik, Magdalena Lemecha, Łukasz Konat, Jerzy Napiórkowski

**Affiliations:** 1The Faculty of Technical Sciences, The University of Warmia and Mazury in Olsztyn, M. Oczapowskiego 11, 10-719 Olsztyn, Poland; 2The Faculty of Mechanical Engineering, Department of Vehicle Engineering, Wroclaw University of Science and Technology, 50-370 Wrocław, Poland

**Keywords:** martensitic steels, abrasive wear resistance, spinning bowl method

## Abstract

This paper presents the results of wear tests of three types of commercial abrasion-resistant steels. The samples, cut from commercially available sheets of metal, were subjected to wear tests to a total friction path of 20,000 m. The tests were provided using the “rotating bowl” method in three types of natural soil masses. The soil moisture and test parameters were kept constant. The tests were carried out in six replications for each material. The testing results indicate that hardness does not determine the resistance to abrasive wear, which is supported by the weight loss results for particular materials. Hardox 600 steel, which is not characterized by the highest hardness, exhibited the lowest weight loss value compared to the other materials in all test soils. For the light soil, the weight loss for Hardox 600 was approx. 1.3 times lower than for Hardox 500 steel and approx. 1.6 times higher than for Hardox Extreme steel. With regards to the medium and heavy soil, the weight losses for Hardox 600 in relation to Hardox 500 steel were approx. 1.7 and 1.6 times lower, respectively, while in relation to Hardox Extreme steel the weight losses were 1.5 and 1.7 times higher, respectively.

## 1. Introduction

High-strength boron-alloyed steels are a large material group including, inter alia, advanced high-strength steels (AHSS), which include TRIPs (transformation-induced plasticity steels), DPs (dual-phase steels), CPs (complex-phase steels), and TWIPs (twinning-induced plasticity steels). The above-mentioned grades, thanks to the application of advanced thermoplastic processing technologies, are characterized by the presence of complex structures that shape their mechanical properties. For example, the TRIP steels, due to the retained austenite content, are strain-hardened through the martensitic transformation of austenite [1]. The ferritic–martensitic structure in dual-phase steels can be obtained through the controlled cooling rate in the Ac1 and Ac3 temperature ranges. Other examples of low-alloy high-strength steels include the HSLA (high-strength low-alloy steel) grade as well as armored steels and abrasion-resistant steels [2,3].

A common characteristic of the above-mentioned material groups is the occurrence of boron microadditives, which considerably increase the steel hardenability. By comparison, their proportion in steel at a level of 0.001–0.003% is equivalent to the content of the following elements: Mn—0.6%; Cr—0.7%; Mo—0.5%; or Ni—1.5% [4]. Boron, due to its atomic magnitude (the smallest substitutional element or the largest interstitial element), inter alia, tends towards segregation on the former austenite grain boundaries, which delays the initiation of phase transformations. Moreover, in order to ensure the highest hardenability, an appropriate combination of the boron content with the remaining elements, i.e., manganese, chromium, and vanadium, whose effects delay diffusion transformations (illustrated with corresponding curves on the TTT diagram), is necessary [5]. Thus, it is possible to obtain a uniform martensitic structure on the cross section of a sheet with a considerable thickness of even up to 130 mm. Due to boron’s considerable affinity for nitrogen and titanium, it is also reasonable to alloy steels with aluminum and titanium, which bind the above gases into intermetallic phases [6]. The available studies also demonstrate that the addition of boron may affect the structure morphology by fragmenting martensite or bainitic ferrite laths [7]. 

The chemical and microstructural properties of steel shaped in this manner are translated into their high strength indices and the maintenance of favorable plastic properties. Examples of structures that require an appropriate combination of the above-mentioned properties are parts exposed to abrasive and impact wear [8], including, e.g., excavator buckets, jaw crushers, and ploughshares. It can be generally concluded that the main factor contributing to an increase in abrasive wear resistance is the hardness level, which increases with an increase in the carbon content. The above statement refers to hard-faced materials [9,10,11], chromium cast iron [12], and martensitic steels [13,14]. It should be noted, however, that selected microstructural parameters (the presence of secondary phases or the former austenite grain size) should also be considered when assessing the abrasive wear resistance, thus affecting the different wear characteristics, even among materials of the same hardness in impact [15,16], abrasive [17,18], and sliding wear conditions [19]. The above relationships affect the degree of surface layer strengthening [20,21] and the impact load resistance [22,23].

The process of reasonable material selection for operating parts and a rational assessment of their operational properties are possible where the course of the wear process is known. Therefore, it appears necessary to properly define the operating conditions of structural assemblies, i.e., the abrasive mass type and the random characteristics of the abrasive itself, such as the moisture content, grain size, and pH value, which affect the tribological testing results that are obtained [24,25].

The wear impact of soil abrasive mass is a complex process in view of the variety of factors affecting its course. Depending on the interaction of the soil mass, the process parameters, and the properties of the working part, wearing can occur in various configurations. The most frequently occurring wear involves loose abrasive particles hitting the surface of a worn material. For compact abrasive masses, soil particles can be treated as fixed grains causing material losses as a result of processes typical for abrasive wear [26,27]. The aspect of the characteristic features of the abrasive mass in modeling abrasive wear has been emphasized by many authors [28,29]. Misra et al. [28] indicate that the wear index increases with an increase in the size of abrasive grains up to 100 μm and loses significance above this size. An abrasive soil mass is a mix of abrasive grains of varied sizes, shapes, and material origins. The material origin of a soil mass has an effect on its wear properties. Woldman et al. [30] presented the varied wear effects of sand particles of varied origin and analyses of the effects of grain size and shape on the wearing process.

Having considered the above-mentioned factors, it should be stated that steels with greater hardness and a higher degree of solid solution hardening can exhibit lower wear resistance in the presence of compact and hard abrasive masses [29,31]. Depending on the abrasive mass interactions and the design and material characteristics of operating parts and operating parameters, wear can occur in different forms. In soil masses containing sandy fractions, wear occurs mainly due to the impact of loose abrasive particles against the surface of an operating part. With regards to compact soils, these particles can additionally occur as fixed grains, which are related to the occurrence of silty fractions in these soils [32].

The aim of the study was to compare the wear processes of various types of commercial wear-resistant steels in different soil conditions and to assess the impact of hardness on the intensity of wear. The obtained results may allow for a rational selection of this type of material for the expected operating conditions.

## 2. Materials and Methods

The subject of the study included three steels with the designations Hardox 500, Hardox 600, and Hardox Extreme, which were supplied directly by the authorized distributor (STAL-HURT, Marciszów, Poland). Specimens with dimensions of 30 × 25 × 10 mm were cut out from sheet metal by the high-energy abrasive waterjet cutting method, i.e., the technology that ensures that the microstructure shaped at the metallurgical processing stage can be preserved. Analyses of the chemical composition were conducted on the cross sections of the sheets using a Leco GDS500A glow discharge emission analyzer (Saint Joseph, MI, USA) with the following parameters applied: U = 1250 V, I = 45 mA, and 99.999% argon, with the obtained results representing the arithmetic means of five measurements. Hardness measurements were conducted using a Zwick/Roel ZHU universal hardness testing machine (Ulm, Germany) by the Brinell method, in accordance with standard PN-EN ISO 6506-92 1:2014-12. A sintered carbide ball with a diameter of 2.5 mm was used, with a load of 187.5 93 kgf (1838.7469 N) applied for 15 s. All measurements were conducted on the specimens that had been subjected to previous microstructure assessments. Metallographic tests were conducted using a Nikon Eclipse MA200 light microscope (Tokyo, Japan) coupled to a Nikon DS-Fi2 digital camera. An analysis of the obtained images was conducted using the NIS Elements software (Nikon Corporation, Tokyo, Japan). The assessment of the microstructure was conducted on specimens etched with a 3% HNO_3_ solution in accordance with PN-H-04503:1961P. In order to reveal the former austenite grain boundaries, the material was tempered at 550 °C for 30 min and then cooled in the air. For that purpose, a Czylok FCF 12SHM/R gas-tight chamber furnace (Jastrzębie-Zdrój, Poland) was used. To reveal the austenite grain boundaries, a Mi7Fe reagent was used (2 g of picric acid, 1 g of sodium alkyl sulfonate, and 100 mL of H_2_O), according to standard ASTM E407. The grain size measurement was conducted in accordance with standard PN-EN ISO 643:2020-07 using ImageJ ver. 1.52. The obtained results represent the arithmetic means of 100 measurements. The photographs showing the microstructures and surfaces subjected to abrasive wear testing were taken with a Keyence VHX 700 digital microscope (KEYENCE INTERNATIONAL (BELGIUM) NV/SA, Mechelen, Belgium) and a Phenom XL electron microscope (Phenom-World, Eindhoven, The Netherlands) using BSE imaging and an accelerating voltage of 15 kV. The photographs were taken with magnifications within a range of 2000–10,000×.

The roughness profile was made using a 3D Formtracer Avant S3000-D profilometer (Mitutoyo, Sakado, Takatsu-ku, Kawasaki, Kanagawa, Japan).

Abrasive wear resistance tests were conducted under laboratory conditions with a “rotating bowl” (a stand of our own construction) test stand that was used to assess wear in loose abrasive materials (Figure 1). The rotating-bowl-type test stand allowed the testing of the abrasive wear of construction materials in the presence of heterogeneous abrasive masses with moisture contents of up to 30%. This allowed the use of natural soils for testing, which is not possible when using a dry sand–rubber wheel test bench (as described in the ASTM G-65 standard) and similar constructions. It was decided to select a rotary bowl stand for testing in order to reflect the operating conditions of the tested materials as much as possible.

The tribological experiment was designed as an active experiment. The objects of the research were samples of three materials taken from commercially available metal sheets. The variables in the experiment were the types of soil masses. Tribological studies were carried out with the rotating bowl method in three types of natural soil masses collected from fields near Olsztyn (Poland). Before the test, the soils were sieved through a sieve with a mesh diameter of 8 mm. In this way, stones were eliminated from the soil masses. During the tests, the load, velocity, and attacking angle of the sample surface on the soil surface were kept constant. Laboratory conditions allowed for the reduction of disturbances in soil moisture and ambient temperature changes. During the tests, the humidity of the soil masses was kept in the range of 11–15%. The tests were carried out with six replications for each material.

The machine bowl was filled with a natural soil mass. Three types of abrasive mass, corresponding to light, medium, and heavy soil, were used. The classification was carried out in accordance with the standard of the Soil Science Society of Poland (PTG), according to PN-EN ISO 14668-2 (2004), and the results are presented in Table 1. The applied research parameters are presented in Table 2.

Mass wear, expressed as weight loss, was adopted as the measure of mass. The masses of the samples were measured every 2000 m using a laboratory balance with an accuracy of 0.0001 g. The mass wear was calculated from the relationship:(1)$$W_{m}=m_{w}-m_{i}$$
where:*W_m_*—wear mass (g);*m_w_*—initial specimen weight before the wear testing (g);*m_i_*—specimen weight after covering a specified friction distance (g).

## 3. Results and Discussion

### 3.1. Chemical and Microstructural Analysis

The carbon contents in the analyzed materials fell within a range of 0.28–0.44% (for Hardox 500, Hardox 600, and Hardox Extreme steel, respectively), which classified them as medium-carbon steels. The obtained results also translated into increasing hardness values. The primary element that increases the hardenability in all grades is boron, and another one is manganese, whose content exhibits a linear decrease with an increase in the steel grade. According to the CEV (carbon equivalent value) formula, this relationship is justified by the increase in the hardenability of materials due to a change in the proportion of the remaining alloying components by the adverse effect on the growth of former austenite grains and by the lowering of the martensite start temperature, Ms. The silicon content was relatively low (approx. 0.2%), which ensured that the plastic properties remained satisfactory. It should be noted that the Hardox 600 and Hardox Extreme steels were also enriched with nickel, which as an element with a different crystal lattice, ensured considerable solid solution hardening and contributed to the lowering of the austenization temperature and the brittle–ductile transition temperature. Moreover, the addition of molybdenum at a level of 0.15% justified the carrying out of the tempering processes by neutralizing the adverse effect of chromium on the temper brittleness, whose contents were similar in all analyzed grades (0.82–0.96%). In all analyzed materials, boron occurred in a concentration typical of low-alloy medium-carbon steels (0.002%). Larger quantities can have an adverse effect on hardenability through the release of iron compounds. The effect of boron is intensified by chromium, nickel, manganese, and molybdenum. Additionally, trace amounts of titanium and aluminum could also be observed. Boron, which exhibits a strong affinity for oxygen and hydrogen, reacts with these elements to form boron nitrides or oxides. The additives of titanium and aluminum bind the above gases into nonmetallic phases so that an appropriate quantity of boron remains dissolved in the matrix, thus guaranteeing the hardenability of the material. It should be noted, however, that (as for Hardox 600 steel) it is reasonable to partially replace titanium with the elements niobium and vanadium because of the possibility of nucleation of micrometric particles contributing to a considerable reduction in fatigue strength and fracture toughness. Since the contents of harmful additives (P and S) were negligible, the test steels were characterized by very high mechanical properties.

The chemical composition and hardness measurement results for the analyzed abrasion-resistant steels (% *w*/*w*) are presented in Table 3.

The microstructural analysis showed that all the materials were characterized by a homogenous fine-lath tempering martensite structure (Figure 2). Moreover, particularly for Hardox 500 steel, areas of hardening martensite, which is a hard and brittle phase, thus exhibiting an adverse effect, e.g., on the resistance to fatigue wear, could be observed locally. Its presence was a result of the method of manufacturing in the steel plant, where a tempering treatment is often omitted, and thus the quenching stresses can only be eliminated in the self-tempering process. In addition, the structure could be determined hierarchically, i.e., through the specification of martensite packages and blocks.

Additional assessment was ensured by the analysis of the former austenite grain size. According to Figure 3, the Hardox 600 steel was characterized by the smallest grain size of 12.2 µm, while the Hardox 500 and Hardox Extreme steels exhibited values of 17.1 and 19.5 µm, respectively. This can be attributed to the increased contents of niobium and aluminum microadditives. The above-mentioned elements tend to form intermetallic phases, which block the migration of grain boundaries at high temperatures, thus allowing a fine-grained structure to be obtained [33,34,35]. Moreover, the microstructure morphology of Hardox Extreme steel was characterized by the presence of abnormal grains larger than 50 µm. A structure formed in this way may affect the results obtained during tribological testing. According to [36], for Hardox 450 steel, the resistance to wear decreased with an increase in the austenization temperature prior to hardening to 1200 °C and the austenite grain size growth to 40 µm. Furthermore, according to [37], steel with a hardness of 500 HBW exhibited lower wear indices where its microstructure comprised equiaxial grains with a size of 14 µm. Similar conclusions were also reached in other studies [14,38]. Based on [39], it can be hypothesized that the largest grain size of the former austenite of 38 GSA steel has an effect on its similar wear indices compared with the less resistant steels TBL PLUS and Creusabro 4800 or with the XAR 600 steel of similar hardness. The carbon content is the highest for 38 GSA steel, and the character of wear itself shows the least favorable character in the form of irregularly arranged grooves and greater portions cut out of the material. It should be noted, however, that the dominant characteristic that affects the wear indices obtained is the mechanical parameters. Alloy additions (e.g., of chromium and nickel) ensure solid solution hardening, with the result that the material shows no tendency towards strain hardening. Thus, the steels were characterized, in morphological terms, with analogous former austenite grain sizes, but in chemical terms, with higher carbon contents and the presence of alloy additions, which affect different strength properties and may exhibit similar wear indices compared to lower-grade steel, in which the weight loss is subject to their assessment.

### 3.2. Abrasive Wear Testing Results

The wear values of the test materials after covering a distance of 20,000 m in particular soils are presented in Figure 4.

A comparison of the wear values for particular materials in test abrasive masses indicated that the highest mass wear values after covering a friction distance of 20,000 m were noted in the heavy soil, followed by the medium and light soils.

In the light soil, Hardox 600 steel was characterized by the lowest wear value. The degree of wear of this steel was approx. 30% lower than that for Hardox Extreme steel and approx. 23% lower than that for Hardox 500 steel.

The analysis of the weight loss results for the test steels in the medium soil enabled the conclusion that the greatest mass losses after covering a friction distance of 20,000 m were noted for Hardox 500 steel (1.6692 g). Hardox 600 steel appeared to be most resistant to wear in this soil, with a wear value of 0.9906 g. As for the heavy soil, Hardox Extreme steel appeared to be the material the least resistant to wear, with a weight loss of more than 2.3 g. It was followed, in terms of the wear value, by Hardox 500 steel, whose wear was approx. two times greater than that for the most resistant Hardox 600 steel.

In order to identify homogeneous groups and determine significant differences in the wear of the test materials in particular soils, a statistical analysis was conducted for the mean values of weight loss over the total friction distance (Table 4, Table 5 and Table 6). For each abrasive mass type, the null hypothesis of a lack of significant differences depending on the test material was adopted. Where the null hypothesis needed to be rejected in favor of an alternative hypothesis, a Duncan test was applied to distinguish homogeneous groups.

Based on the obtained results, it can be concluded that there were statistically significant differences in wear for the tested construction materials, irrespective of the abrasive medium type. This fact was evidenced by the hardness of test materials and the associated plasticity as well as the mechanical properties associated with the alloy additions.

The analysis of the wear processes was conducted based on images of the surfaces after the conducted tribological tests (Figure 5, Figure 6, Figure 7, Figure 8 and Figure 9). They confirm the material wear values obtained from testing in particular abrasive masses. The link between these relationships can be seen in the grain size distributions in the soils. In the medium and heavy soils, there was less of the loosely interconnected sandy fraction than in the light soil and considerably more of the silty fractions and dust. These increased the soil cohesion by reducing the degrees of freedom of loose sandy grains, whose hardness reaches up to 1300 HV10 [25]. With increases in the contents of silty fractions, the test material friction process changed fundamentally, as illustrated in Figure 5.

The main wear pattern for Hardox 500 steel in the light soil (Figure 5a) was scratching and individual ridging traces with the partial removal of material from the groove. Spots of material losses also occurred due to chipping caused by microfatigue resulting from the cyclic impact of contact stresses in the surface layer due to the impact of sand grains under load. This wear pattern can be classified as fatigue wear due to spalling. Plastic deformations, which are the first stage of chipping wear, could also be seen.

On the surfaces of the Hardox 500 steel worn in the medium (Figure 5d) and heavy soil (Figure 5g), the wear types were similar to those noted in the light soil. However, the damage to the surface was more intense compared to the light soil. It was dominated by intense patterns of mechanical impact through ridging and microcutting. This was linked to the impact on the material of grain sands fixed by the silty fractions found in these soils. The processes of such a wear pattern in the heavy soil were so dominant that wear by spalling or scratching was essentially nonexistent.

On the surface of Hardox 600 steel worn in light soil (Figure 5b), the main patterns included scratching, ridging, and fatigue wear of the material due to the impact of loose abrasive grains on the surface. It is significant that, despite the increased hardness of the steel in relation to Hardox 500 steel, individual microcutting traces (Figure 5d) occurred as well. These were due to the actual positioning of an abrasive grain’s sharp edge in relation to the material being abraded. Their effect on the wear intensity was negligible, as the wear of Hardox 600 steel, as compared to Hardox 500 steel in the light soil, was lower by 33%. In the medium soil, similar wear patterns occurred, but the ridging intensity, compared to that in the light soil, was considerably higher.

The appearance of the surface of Hardox 600 worn in heavy soil (Figure 5h) indicated the dominance of chipping, ridging, and microcutting in the wear process. Chipping was due to the fact that in this tribological combination, in addition to the typical mechanical wear, classical mechanochemical wear occurred. The ploughing process partially removed material and caused the plastic deformation of the edge of the furrow (pushing material out). The pushed-out material was weakly bonded to the sample surface, which facilitated its removal under the influence of the next abrasive grains. This wear process was promoted by the high contents of silty and dusty fractions (84.18%) and the low sandy fraction content (15.82%). The noted wear by ridging and microcutting resulted from the cutting impact of sand grains fixed in silt and dust.

On the surface of Hardox Extreme steel worn in light soil (Figure 5c), ridging traces were primarily visible. These grooves were considerably wider and deeper than those resulting from the wearing of the other steels in this abrasive mass. Plastic deformations and material tear-ups due to the impact of loose abrasive grains could also be noted. The appearance of the surface worn in the medium soil indicated wear mechanisms typical of this abrasive mass type (Figure 5f). Ridging and microcutting could be seen. These traces were considerably more intense compared to the other materials worn in this type of abrasive mass. However, for the heavy soil, mainly wide and deep grooves could be seen on the surface, which reflected the weight loss of this material (Figure 5i).

The characterization of the worn surfaces using 3D surface profilometry indicated the pronounced differences in surface morphology, i.e., the degree of surface plastic deformation, between the specimens tested in different abrasive masses (Figure 6). To evaluate the roughness of the surfaces shown by the 3D profile, the following parameters were used: S_k_—core roughness depth; S_kv_—reduced dale height; and S_pv_—reduced peak height, determined for a sampling density of 10 µm, corresponding to the residual surface after normalization with a second-order polynomial.

The obtained values showed that the furrows formed as a result of the abrasive wear of the tested materials were the largest for Hardox 500 steel. For this steel grade, the highest values of the S_k_ and S_vk_ parameters were obtained in all abrasive weights. It should be noted that the highest values of the S_k_ parameter for Hardox 500 steel were observed in light soil, which contains the highest amount of large abrasive particles (sand fractions). These grains caused the formation of deep scratches in the material of the lowest hardness. The value of the S_pk_ parameter is related to the plastic deformation of the material at the edges of the grooves and the exposure of hard phases of the martensitic microstructure. The values of the measured roughness parameters correlated with the microscopic images of the worn surfaces of the samples.

The surface descriptions presented above show the complex character of the soil excitation impact. The hardness of the material had no unambiguous effect on the occurrence of the specified wear patterns, particularly in terms of the destructive impact intensity. The occurrence of the elemental wear phenomena was primarily determined by the soil fraction content. It should be stressed, however, that even for homogeneous structures, it was only possible to identify the dominant wear pattern and not several of them. This was determined by the random arrangement of the soil grains in relation to the wearing part. The frequency and intensity of the occurrence of particular wear patterns on the friction surface were reflected in the total mass wear, which enabled the quantification of the materials used to manufacture tools used in the soil.

Additional information on the course of wear was provided by a microstructural assessment of the cross sections of the surfaces subjected to testing. For the Hardox 500 and Hardox 600 steels (Figure 7 and Figure 8), the surfaces were relatively smooth, with no sharp indentations or tear-ups. The deformations located very close to the surface and under cavities were indicative of the plastic yielding of martensite blocks and the strain hardening of the surface layer. In addition, the considerable changes in the height were a result of the plastic pushing of the material, manifested by the occurrence of grooves. The above-mentioned wear mechanisms were not observed for Hardox Extreme steel, as the pronounced surface unevenness and sharp edges were indicative of material cut-out (Figure 9). Furthermore, the depth and width of the abrasive agent impact traces increased with a change in the abrasive mass type.

## 4. Conclusions

Based on the testing that was conducted, the following conclusions can be formulated:The Hardox steel microstructure is characterized by the homogeneous tempering martensite morphology, with areas of hardening martensite for Hardox 500 steel. The analysis of the former austenite grain size showed that all the steels should be classified as fine-grained. The Hardox 600 steel was characterized by the smallest grain size of 12.2 µm, while the Hardox 500 and Hardox Extreme steels exhibited values of 17.1 and 19.5 µm, respectively. The above differences may have been due to the higher content of alloy microadditions (niobium and aluminum) in Hardox 600 steel, which tend to form intermetallic phases that block the grain boundary migration and thus ensure that the fine-grained structure is preserved. Furthermore, at the metallurgical processing stage, the initial temperature for the hardening processes could have been too high.The lowest weight loss value compared to other materials in all soil types was noted for Hardox 600 steel, which was characterized by the smallest former austenite grain size. For the light soil, it was lower by approx. 1.3 times compared to Hardox 500 steel and approx. 1.6 times higher than for Hardox Extreme steel. With regards to the medium and heavy soil, these differences were, in relation to Hardox 500 steel, approx. 1.7 and 1.6 times, respectively, while in relation to Hardox Extreme steel, these differences were 1.5 and 1.7 times.The highest intensity of Hardox Extreme steel wear was a result of the low plasticity of the material, which made microcutting the main wear mechanism. The material showed no tendency towards strain hardening, which was due to the highest carbon content and the degree of solid solution hardening. For the Hardox 500 and Hardox 600 steels, the plastic yielding of martensite blocks occurred along the direction of the abrasive mass action. It can also be assumed that the greater size of the former austenite grain contributed to a reduction in the abrasion wear resistance.The different chemical and morphological characteristics affected the complex and diverse wear patterns of this material group. Wear marks were equally intense in all steels. However, for the Hardox 600 and Hardox 500 steels, grooves were dominant, as the abrasive grains caused the pushing of the material and its subsequent removal. On the other hand, for Hardox Extreme steel, the dominant wear pattern was microcutting.The analyzed Hardox 500, Hardox 600, and Hardox Extreme steels fulfil the requirements set for them, as they are characterized by high hardness (which increases with an increase in the steel grade). However, further research is required to determine the possibility of using these steels for machinery operating parts.

## Figures and Tables

**Figure 1 materials-15-07622-f001:**
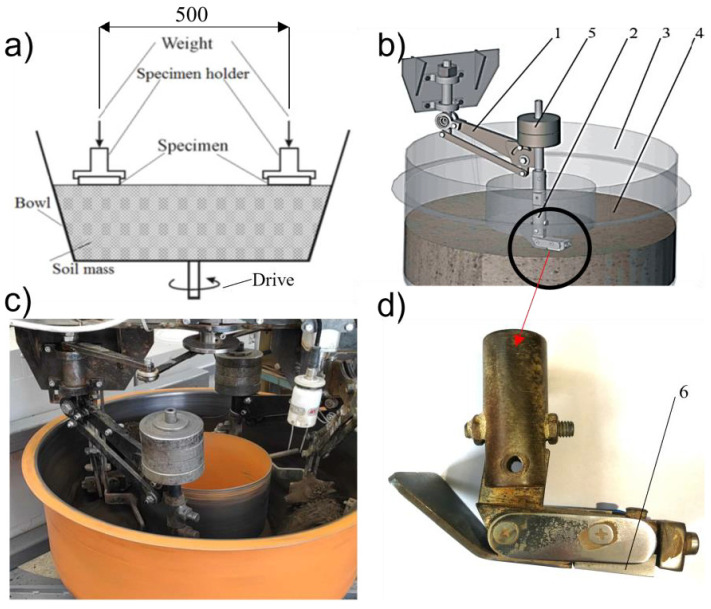
A “rotating bowl” test stand: (**a**) the principle of operation; (**b**) a scheme of one working section: 1—swingarm, 2—specimen holder, 3—bowl with abrasive mass 4, 5—weight, 6—specimen; (**c**) general view; (**d**) specimen holder.

**Figure 2 materials-15-07622-f002:**
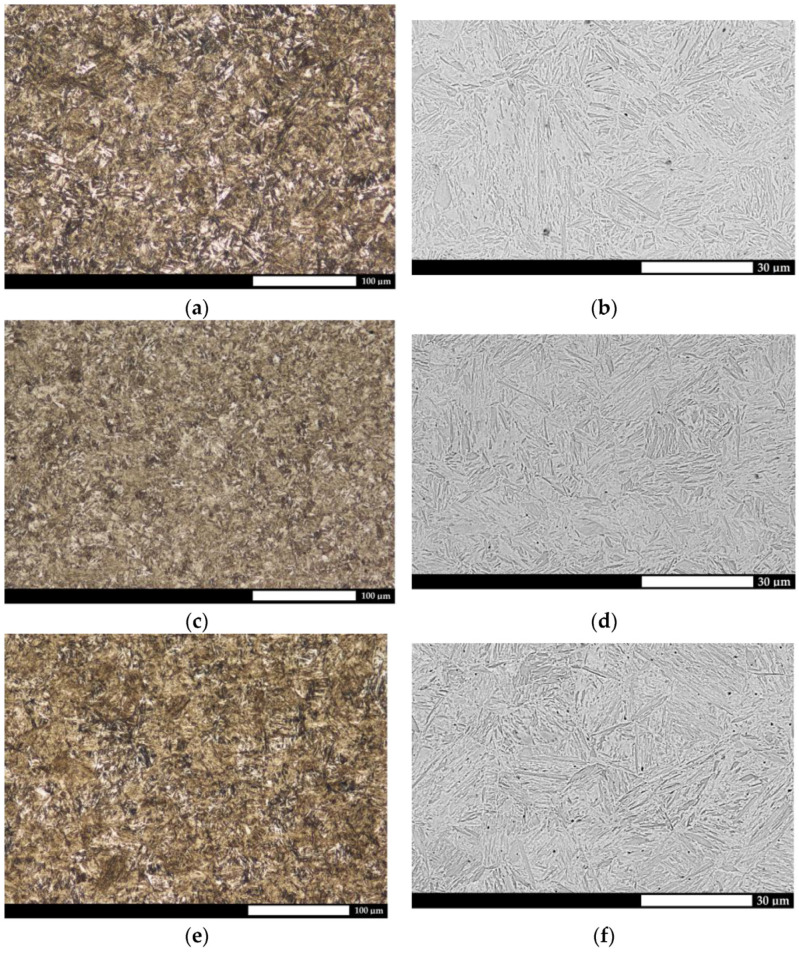
Microstructures of the analyzed abrasion-resistant steels. (**a**,**b**) Hardox 500, (**c**,**d**) Hardox 600, (**e**,**f**) Hardox Extreme. Light and electron microscopy. Etched with Mi1Fe (Nital).

**Figure 3 materials-15-07622-f003:**
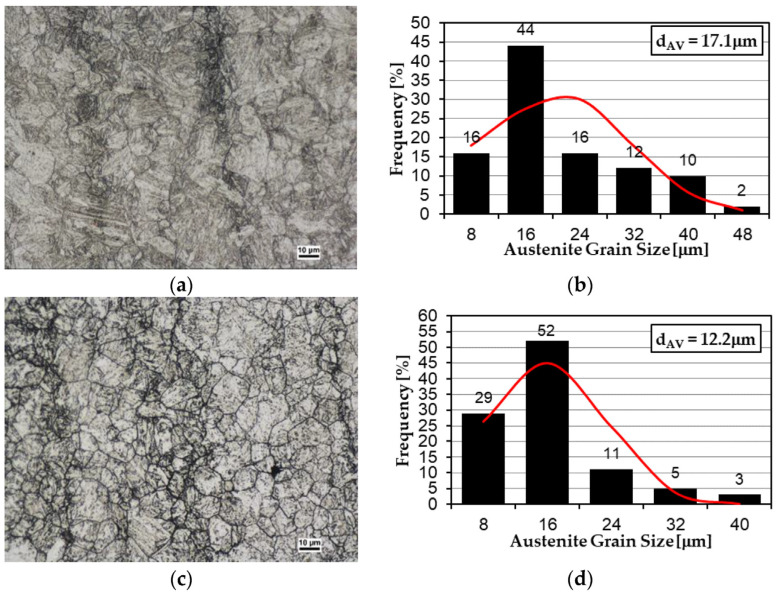
Microstructures of the analyzed abrasion-resistant steels with revealed former austenite grain boundaries. (**a**,**b**) Hardox 500, (**c**,**d**) Hardox 600, (**e**,**f**) Hardox Extreme. Light microscopy. Etched with Mi17Fe (Pikral).

**Figure 4 materials-15-07622-f004:**
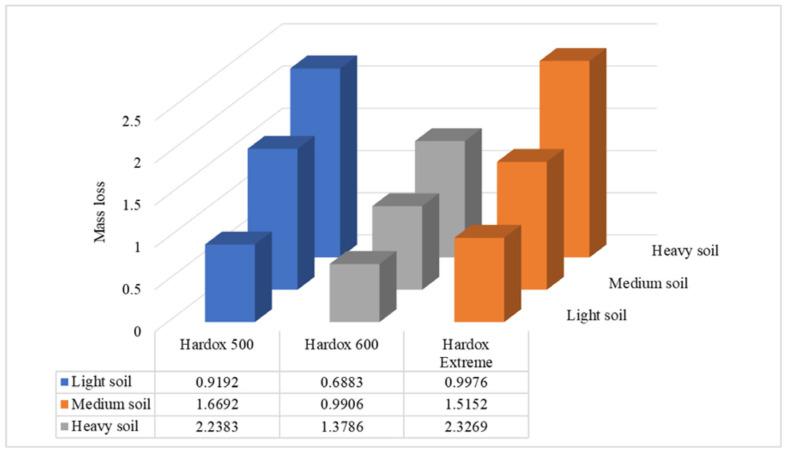
Total weight losses for test materials in particular soil types.

**Figure 5 materials-15-07622-f005:**
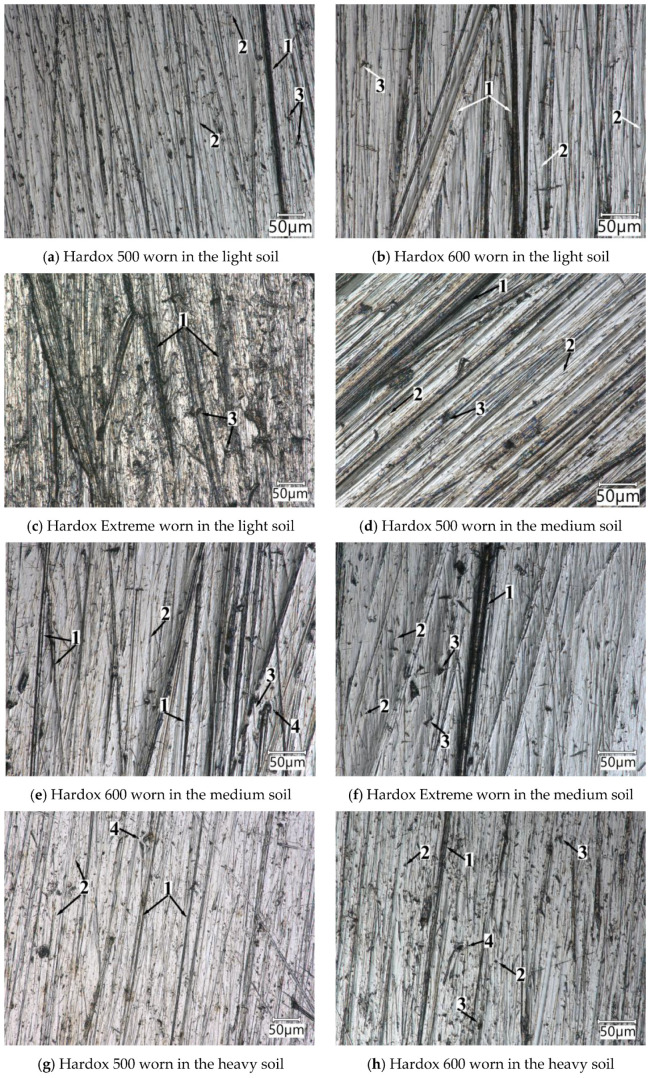
The surfaces of the tested steels worn in the different soil masses: 1—ploughing, 2—microcutting, 3—chippings, 4—plastic deformation. Light optical microscopy.

**Figure 6 materials-15-07622-f006:**
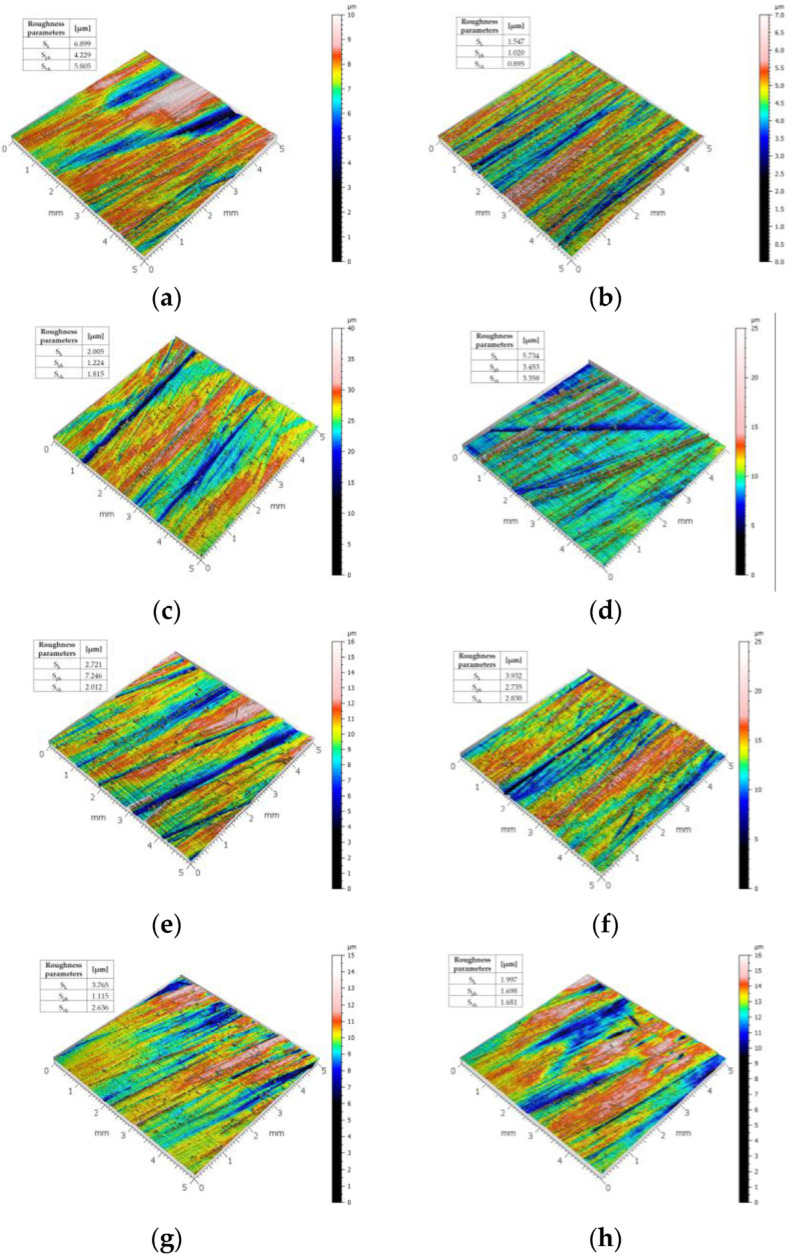
The 3D roughness profiles of the tested materials’ surfaces: (**a**) Hardox 500 worn in light soil, (**b**) Hardox 600 worn in light soil, (**c**) Hardox Extreme worn in light soil, (**d**) Hardox 500 worn in medium soil, (**e**) Hardox 600 worn in medium soil, (**f**) Hardox Extreme worn in medium soil, (**g**) Hardox 500 worn in heavy soil, (**h**) Hardox 600 worn in heavy soil, (**i**) Hardox Extreme worn in heavy soil.

**Figure 7 materials-15-07622-f007:**
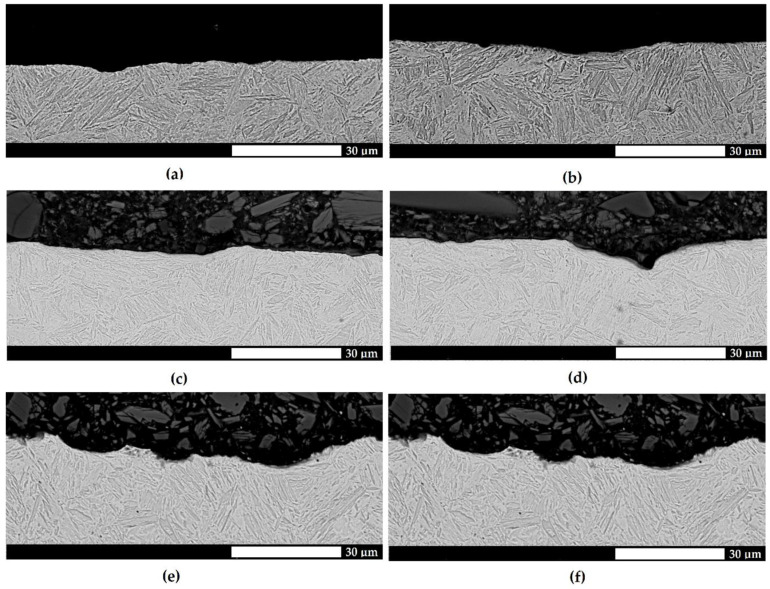
Microstructures of the surfaces subjected to testing for the abrasive wear of Hardox 500 steel in soil on a cross section transverse to the direction of the abrasive action. (**a**,**b**) light soil; (**c**,**d**) medium soil; (**e**,**f**) heavy soil. The surface is relatively smooth, and the distortion of martensite blocks can be observed. Electron microscopy, etched with Mi1Fe (Nital).

**Figure 8 materials-15-07622-f008:**
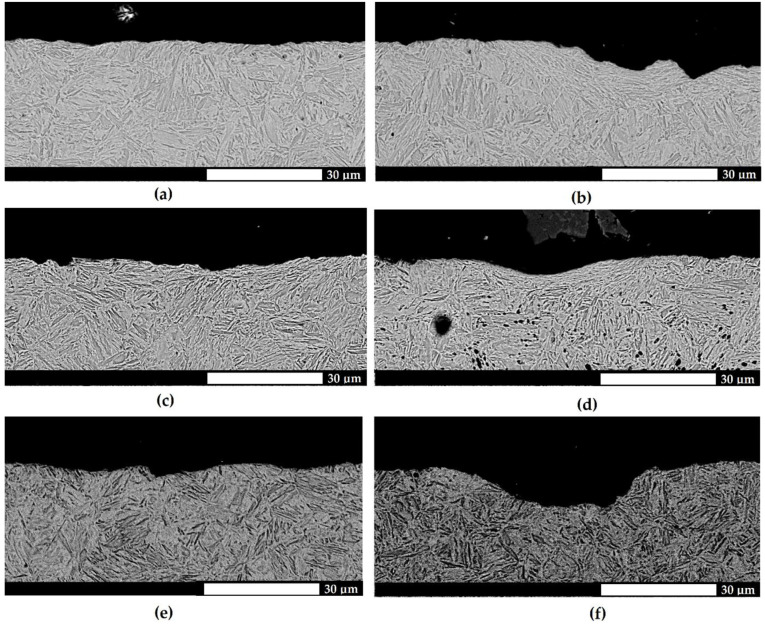
Microstructures of the surfaces subjected to testing for the abrasive wear of Hardox 600 steel on a cross section transverse to the direction of the abrasive action. (**a**,**b**) light soil; (**c**,**d**) medium soil; (**e**,**f**) heavy soil. The surface is relatively smooth, and the distortion of martensite blocks can be observed. Electron microscopy, etched with Mi1Fe (Nital).

**Figure 9 materials-15-07622-f009:**
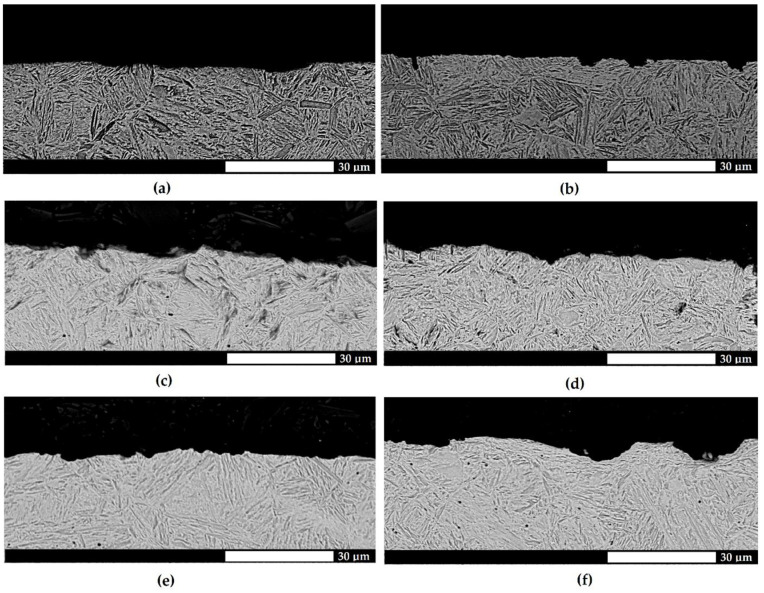
Microstructures of the surfaces subjected to testing for the abrasive wear of Hardox Extreme steel on a cross section transverse to the direction of the abrasive action. (**a**,**b**) light soil; (**c**,**d**) medium soil; (**e**,**f**) heavy soil. Characteristic sharp edges are seen with no distortion of martensite blocks. Electron microscopy, etched with Mi1Fe (Nital).

**Table 1 materials-15-07622-t001:** Abrasive soil mass characteristics.

Soil Mass Type	Granulometric Group	Fraction Content (%)			Moisture	Soil Acidity
		Sand 0.05–2.00 mm	Dust 0.002–0.05 mm	Loam <0.002 mm	Vol (%)	pH
Light	Loamy sand	67.17	31.03	1.80	11	6.6–7/ neutral
Medium	Light clay	62.98	34.01	2.97	13	6.6–7
Heavy	Loam	15.82	76.71	7.47	15	6.6–7

**Table 2 materials-15-07622-t002:** The testing parameters.

The Testing Parameters		
Sliding Speed (m∙s^−1^)	Specimen Load (N)	Total Friction Distance (m)
1.7	49	20,000

**Table 3 materials-15-07622-t003:** Chemical composition and hardness measurement results of the analyzed wear-resistant steels (wt. %).

	Hardox 500	Hardox 600	Hardox Extreme
C	0.28	0.38	0.44
Mn	1.15	0.85	0.55
Si	0.25	0.21	0.19
P	0.007	0.01	0.007
S	0.003	-	0.002
Cr	0.96	0.82	0.84
Ni	0.06	1.10	2.05
Mo	0.03	0.14	0.15
V	0.015	0.01	0.011
Cu	0.029	0.032	0.022
Al	0.063	0.10	0.045
Ti	0.002	0.002	0.004
Nb	-	0.02	-
Co	0.028	0.002	0.01
B	0.0006	0.0021	0.0019
HBW	487 ± 2	596 ± 9	618 ± 6

**Table 4 materials-15-07622-t004:** The results of the statistical analysis of the differences between the average wear values for particular materials in the light soil.

Subclass No.	Duncan Test: Light Soil, Variable Homogeneous Groups, Alpha = 0.05000 Error: Intergroup MS = 0.00038, df = 15,000				
	Material	Average after Distance 20,000 m	1	2	3
2	Hardox 600	0.6882	****		
1	Hardox 500	0.9191		****	
3	Hardox Extreme	0.9975			****

****—illustrates a statistically significant difference with *p* < 0.05 between the materials.

**Table 5 materials-15-07622-t005:** The results of the statistical analysis of the differences between the average wear values for particular materials in the medium soil.

Subclass No.	Duncan Test: Medium Soil, Variable Homogeneous Groups, Alpha = 0.05000 Error: Intergroup MS = 0.00038, df = 15,000				
	Material	Average after Distance 20,000 m	1	2	3
2	Hardox 600	0.9905	****		
3	Hardox Extreme	1.5152		****	
1	Hardox 500	1.6692			****

****—illustrates a statistically significant difference with *p* < 0.05 between the materials.

**Table 6 materials-15-07622-t006:** The results of the statistical analysis of the differences between the average wear values for particular materials in the heavy soil.

Subclass No.	Duncan Test: Heavy Soil, Variable Homogeneous Groups, Alpha = 0.05000 Error: Intergroup MS = 0.00038, df = 15,000				
	Material	Average after Distance 20,000 m	1	2	3
2	Hardox 600	1.3785	****		
1	Hardox 500	2.2382		****	
3	Hardox Extreme	2.3268			****

****—illustrates a statistically significant difference with *p* < 0.05 between the materials.

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
