# Peer review of "Analysis of Wear Properties of Hardox Steels in Different Soil Conditions"

_materials, 2022, doi:10.3390/ma15217622_

Round 1

Reviewer 1 Report

Authors have reported different abrasive behavior of Hardox steel in varied Soil using unique rotating bowl experiments. Nevertheless, the results provided in the manuscript don't appear to support the Worn surface images' behavior. The authors are suggested to support their results with some of the defined tribological tests such as scratch, and 3D roughness profile, Since wear, can behaviour cant be simply defined by mass loss. The manuscript needs to be reorganized to support the comparative performance of steel for better comprehension ability. 

Some of the other Comments are :

1.     Authors need to support the statement with literature on "increased content of niobium and aluminium micro-187 additives. The above-mentioned elements tend to form intermetallic phases, which block 188 the migration of grain boundaries at high temperatures, thus allowing a fine-grained structure to be obtained"

2.     I am not sure how the authors came up with "4 – plastic deformation " in Fig 5. I am not sure if marked areas correspond to plastic deformation. Please rectify them figure or support it with a high-resolution image.

3.     It would be great if authors could compare the light microscopy image of worn surfaces of different steels in one figure for one soil at a time. It will be easier to comprehend the comparison of abrasive wear resistance. Please rearrange the Figures and rewrite in terms in terms of comparison of Steel in similar soils condition 

4.     The image shown in Figure 6. The surface of Hardox 600 steel (a) appears to have more abrasive lining than in Figure 5. The surface of Hardox 500 steel worn in the light soil (a) medium soil. Despite a 33% loss in opposite mass."Can authors support this opposite behavior.

5.     I think the roughness profile of the worn surface can be included it may help in decoding the abrasive behavior of Hardo steel. It would be great if included.

6.     It would be great if you can support the abrasive wear resistance of different steels through a scratch test at the same loading parameter as uniform

7.     Some lines in the manuscript are very confusing and need appropriate modification such as What does the author want to imply through these lines "The structures weakly bonded to the surface layer are then torn out in the mechanical wear process. This wear occurs when the formation of secondary structures proceeds more slowly than mechanical wear"

Author Response

Thank you for your in-depth review of our manuscript. Your comments allowed us to improve our work. All explanations are in the attached file.

Reviewer 2 Report

The manuscript is relevant in terms of industrial applications. However, I have major comments to enhance the quality of the research work.

1)     Please modify the title so that it should reflect the originality of the manuscript.

2)     In the abstract, please write a few sentences on the methodology of the experiment.

3)     Please mention the type or property of soil used in the abstract.

4)     Also, I did not see any discussion on soil type and hardness properties in the introduction section.

5)     Line: 79. Please mention a clear work objective along with a short description of the methodology.

6)     Did you mention the chemical composition of all the Hardox steel samples?

7)     Mention the source of procurement of Hardox steel samples in the Materials and method section.

8)     Please add all machine names/brand/ year etc. I see some experimental machines are not identified.

9)     Can you provide the properties of soil in tables?

10)          Please provide the design of the experiments.

11)          Why did you choose the rotary bowl stand? Could you provide some arguments?

12)          Line 121 to 130 should be provided in the table.

13)          Line 198: Are there any verified results for this argument?

14)          Check Tables 3 to 4 for any missing value.

15)          Check spelling mistakes.

Author Response

(The authors gave the same response as above.)

Reviewer 3 Report

This document offers hardness tests on Hardox steel abrasive wear. Hardox 500, 600, and Extreme were tested. "Rotating bowl" experiments were done with light, medium, and heavy soils. The weight loss data for some materials reflect the testing results that hardness does not predict abrasion resistance. Hardox 600 steel, which isn't the hardest, had the lowest weight loss in all test soils. In light soil, Hardox 600 had 1.3 times less weight loss than Hardox 500 and 1.6 times more than Hardox Extreme. In medium and heavy soil, Hardox 600 weight losses were 1.7 and 1.6 times lower than Hardox 500, but 1.5 and 1.7 times greater than Hardox Extreme.

I recommend this paper for publication after a moderate correction as follows:

1. In the Introduction, in the fourth paragraph, references should be discussed more clearly than in a series of citations.

2. At the end of the introduction, the authors should describe the structure of the paper so that the transitions of sections are smooth. Example "The rest of this paper is structured as follows, and so on".

3. In the entirety of Section 2, very few references are cited. Therefore, the author should find reliable documents for reference.

4. The information shown in Table 2 need to include citations from reliable sources in the area labeled "Results and discussion."

5. In the Abstact and conclusion section, the authors should further discuss the significance and application of this study in practice.

6. Some errors in English can be corrected as follows:

the results of testing-> experimental results

characterised -> characterized 

soils mass -> soil masses

denability->deniability?

and son on in the whole manuscript.  Authors should use English editing tools to improve manuscripts.

Author Response

(The authors gave the same response as above.)

Round 2

Reviewer 1 Report

The authors have significantly improved the manuscript and have incorporated most of the Queries. I appreciate the authors have included the  3D profile images, however, have not mentioned their relevant data (Ra, Rq)

1. The Authors have not correlated the roughness Ra and Rq factor of 3D roughness images in the text and have barely mentioned anything how it (Fig 6) correlates with wear behavior with the wear behavior of Steel if any. 

Please include Ra and Rq in Figure 6. The 3D roughness profile of the tested materials surfaces. 

Author Response

Thank You for Your suggestions. The reply is in the attached file.

Reviewer 2 Report

Thanks for your response. I would suggest the following title for the research paper

Analysis of wear properties of Hardox steels in differential soil conditions

Author Response

Thank You for Your suggestion. All Authors accepted the proposed title of the manuscript.